# Effect of immortal time bias on the association between immune-related adverse events and oncological outcomes following immune checkpoint inhibitors therapy for head and neck squamous cell carcinoma

**Koichi Tamura, Yukinori Takenaka©\*, Kiyohito Hosokawa, Takashi Sato, Takeshi Tsuda, Hirotaka Eguchi, Masami Suzuki, Takahito Fukusumi, Motoyuki Suzuki, Hidenori Inohara**

Department of Otorhinolaryngology-Head and Neck Surgery, Osaka University Graduate School of Medicine, Suita, Osaka, Japan

\* ytakenaka@ent.med.osaka-u.ac.jp

## Abstract

Immune checkpoint inhibitors (ICIs) are pharmacological agents indicated for recurrent and metastatic head and neck squamous cell carcinoma (HNCSCC). Immune-related adverse events (irAEs) have been reported as predictors of therapeutic response to ICIs. However, previous studies have not adequately addressed the immortal time bias. Therefore, we aimed to investigate the association between the onset of irAEs and oncological outcomes, accounting for immortal time bias. We conducted a retrospective study involving 130 patients with HNSCC who were treated with ICIs. The objective response, progression-free survival (PFS), and overall survival (OS) were assessed using logistic regression analysis, the Kaplan–Meier method, and the Cox proportional hazard (PH) model. The immortal time bias was considered using a landmark analysis and an extended Cox (EC) model. The odds ratios for response and disease control were smaller in the landmark than in the naïve analyses. In the landmark analysis, the 1-year PFS rates were 47.6% and 27.2% for irAE+ and irAE- patients, respectively (p = 0.049), and the 1-year OS rates were 85.7% and 66.5%, respectively (p = 0.006). Regarding PFS, the adjusted HRs for irAEs were 0.49 (95% confidence interval (CI) 0.28–0.85) in the PH analysis and 0.75 (95% CI 0.40–1.40) in the EC analysis. As for OS, the adjusted HRs for irAEs were 0.36 (95% CI 0.19–0.66) in the PH analysis and 0.51 (95% CI 0.27–0.95) in the EC analysis. IrAEs were an independent prognostic factor for OS but not PFS. Without considering the immortal time bias, the association between irAEs and oncologic outcomes in patients with HNSCC treated with ICIs was overestimated. Therefore, the balance between the benefits and risks of ICI therapy must be carefully weighed in clinical settings.

**Data Availability Statement:** All relevant data are within the paper and its Supporting information files.

**Funding:** This work was supported by JSPS KAKENHI; Grant Number JP24K12647. The funder had no role in study design, data collection, analysis, decision to publish, or preparation of the manuscript.

**Competing interests:** The authors have declared that no competing interests exist.

## Introduction

The prognosis of recurrent and/or metastatic (R/M) head and neck squamous cell carcinoma (HNSCC) is poor, with a median crude survival of less than 1.5 years even with systemic chemotherapy [1]. Currently, immune checkpoint inhibitors (ICIs), pembrolizumab and nivolumab, have replaced chemotherapy as the mainstay of treatment [2]. Although long-term survival is observed in some patients, the overall prognosis remains largely unchanged. Despite programmed death-ligand 1 (PD-L1) expression being used as a biomarker for predicting treatment response in anti-programmed death-1 (PD-1) immunotherapy, its usefulness is limited [2].

Unlike cytotoxic anticancer agents, ICIs cause immune-related adverse events (irAEs), the timing and organs of onset of which are difficult to predict [3]. The median time to the onset of fatal irAEs is 14–40 days and the time of irAE onset varies widely among patients [3]. Reports have shown an association between the occurrence of irAEs and therapeutic efficacy of ICIs in several carcinomas, including head and neck cancer [4, 5]. Patients who experience irAEs generally experience long-term survival. In other words, patients who die early are less likely to develop irAEs. Therefore, the survival period of patients with irAEs is longer than that of patients without irAEs, which is referred to as immortal time bias. Unfortunately, most previous studies did not consider immortal bias in their analyses or relied solely on landmark analyses, thus failing to adequately evaluate the association between irAEs and the therapeutic effect of ICIs [4, 6]. Furthermore, these studies rarely incorporate extended Cox models with irAEs as time-varying covariates [4–6].

This study aimed to investigate the true association between ICI outcomes and irAEs in patients with R/M HNSCC, addressing immortal time bias by utilizing both landmark analysis and an extended Cox model.

## Patients and methods

The study protocol was approved by the Institutional Review Board (IRB) of Osaka University Hospital (approval number: 16329–3) and was conducted in accordance with the 1964 Declaration of Helsinki and its later amendments. Given the retrospective nature of the study, the need for informed consent was waived by the IRB, as permitted by the ethical guidelines for epidemiological research published by the Ministry of Health, Labour, and Welfare of Japan.

### Patients and data extraction

The inclusion criteria were: (1) histologically or cytologically confirmed HNSCC and (2) treatment with nivolumab or pembrolizumab for R/M HNSCC at the Department of Otorhinolaryngology-Head and Neck Surgery, Osaka University Hospital, between May 2017 and May 2023. The exclusion criteria were: (1) history of ICI therapy for other malignancies, and (2) insufficient clinical data. A retrospective chart review was performed for records that met the abovementioned criteria. Data on sex, age, Eastern Cooperative Oncology Group Performance Status (PS), primary tumor site, human papillomavirus (HPV) status, ICI and chemotherapy regimens, treatment line for R/M disease, number of ICI administration cycles, and AEs were collected. Two authors (K. T. and Y.T.) accessed the medical records to collect data on May 20~24th, 2024. The collected data were anonymized for the protection of personal information. IrAEs were defined as any AEs considered immune related by a physician. These included AEs requiring corticosteroid treatment, and endocrine AEs requiring hormone replacement therapy. Thyroid dysfunction is often observed in patients with R/M HNSCC because of prior surgery or radiation therapy. Therefore, thyroid dysfunction was determined to be an irAE when considerable changes in thyroid hormone levels were observed after initiation of ICI therapy.

### Oncologic outcomes

The objective response was evaluated using the Response Evaluation Criteria in Solid Tumours (version 1.1 [7]). The response rate (RR) was defined as the percentage of patients who achieved complete response (CR) or partial response (PR). Disease control rate (DCR) was defined as the percentage of patients who achieved CR, PR, or stable disease (SD). Overall survival (OS) was defined as the time from the initiation of ICI treatment to death from any cause. Progression free survival (PFS) was defined as the time from the initiation of ICI treatment to disease progression or death from any cause.

### Statistical analysis

The odds ratio (OR) for objective responses was calculated using a logistic regression model. Survival was estimated using the Kaplan–Meier method and compared using the log-rank test. Landmark analyses and an extended Cox (EC) model were used to correct for immortal time bias. In the landmark analysis, a time point (landmark) is designated and patients who survive longer than the landmark are analyzed. The EC model allows a time-dependent variable to be used as a covariate, and can thus consider the timing of irAE onset. The hazard ratio (HR) for death was calculated using Cox proportional hazard (PH) and EC models. A probability ($p$) value of $<0.05$ was considered statistically significant. These analyses were performed using JMP version 16 statistical software (SAS Institute Japan, Tokyo, Japan) and EZR (Saitama Medical Center, Jichi Medical University, Saitama, Japan), a graphical user interface for R (R Foundation for Statistical Computing, Vienna, Austria).

## Results

### Patient characteristics

After applying the inclusion and exclusion criteria, 130 patients were included in this study (S1 File). The clinicopathological characteristics of the patients are summarized in Table 1. Male patients comprised 80.8% of the cohort, and 60.8% of the patients were $>65$ years of age at the start of ICI treatment. PS was three and two in 3.1% and 10.8% of the patients, respectively. HPV-associated oropharyngeal cancer accounted for 6.2% of the patients. ICI monotherapy was administered to 70.8% of the patients, while the rest received a combination of ICI therapy with cytotoxic drugs. ICIs were administered as the first-line treatment for R/M disease in 70.8% of the patients. Most of the remaining patients received platinum-based chemotherapy in combination with cetuximab, either with docetaxel or fluorouracil, as prior therapy. No significant differences were observed in any variables between patients with and without irAEs. The median follow-up period for surviving patients was 33.4 months.

### IrAEs

IrAEs were observed in 16.1% of patients. The most common irAEs were thyroid dysfunction (nine cases), adrenal insufficiency (three cases), colitis, liver dysfunction, and skin disorders (two cases each). The median time from initiation of ICI therapy to irAE onset was 3.8 months (range: 0.99–29.3).

### Objective response and irAEs

Of the 130 patients, 119 were evaluated for an objective response. The remaining 11 patients were not evaluated for objective response due to death or deteriorated physical conditions. Overall, CR, PR, SD, and PD were observed in 12.6%, 26.9%, 16.0%, and 44.5% of the patients, respectively. The RR and DCR were 39.5% and 55.5%, respectively.

**Table 1. Patient characteristics.**

| | Total (n = 130) | | Patients with irAEs (n = 21) | | Patients without irAEs (n = 109) | | P value |
|---|---|---|---|---|---|---|---|
| | No. | % | | | | | |
| Sex | | | | | | | |
| Male | 105 | 80.8 | 18 | 85.7 | 87 | 79.8 | 0.518 |
| Female | 25 | 19.2 | 3 | 14.3 | 22 | 20.2 | |
| Age, years | | | | | | | |
| Median | 68 | | 71 | | 68 | | 0.451 |
| Range | 21–90 | | (31–84) | | (21–90) | | |
| Performance Status | | | | | | | |
| 0 | 61 | 46.9 | 13 | 61.9 | 48 | 44.0 | 0.322 |
| 1 | 51 | 39.2 | 6 | 28.6 | 45 | 41.3 | |
| $\geq$2 | 18 | 13.8 | 2 | 9.5 | 16 | 14.7 | |
| Primary site | | | | | | | 0.221 |
| Oral cavity | 20 | 15.4 | 2 | 9.8 | 18 | 16.5 | |
| Nasopharynx | 16 | 12.3 | 1 | 4.8 | 15 | 13.8 | |
| Oropharynx | 25 | 19.3 | 4 | 19.1 | 21 | 19.2 | |
| HPV+ | 8 | 6.2 | 0 | 0 | 8 | 7.3 | |
| HPV- | 17 | 13.1 | 4 | 19.1 | 13 | 11.9 | |
| Hypopharynx | 33 | 25.4 | 6 | 28.6 | 27 | 24.8 | |
| Larynx | 11 | 8.5 | 4 | 19.1 | 7 | 6.4 | |
| Other | 25 | 19.2 | 4 | 19.1 | 21 | 19.3 | |
| Regimen | | | | | | | 0.623 |
| Nivolumab | 71 | 54.6 | 12 | 57.1 | 59 | 54.1 | |
| Pembrolizumab | 21 | 16.2 | 2 | 9.5 | 19 | 17.4 | |
| Pembrolizumab+FP | 38 | 29.2 | 7 | 33.3 | 31 | 28.4 | |
| Lines of treatment | | | | | | | 0.942 |
| 1st line | 92 | 70.8 | 15 | 71.4 | 77 | 70.6 | |
| 2nd or later lines | 38 | 29.2 | 6 | 28.6 | 32 | 29.4 | |

Abbreviations: HPV, human papillomavirus, FP, fluorouracil + cisplatin or carboplatin, IQR, interquartile range

The RR in patients who experienced irAEs and those who did not were 66.7% and 33.7%, respectively. The OR for response was 3.94. Landmark analyses were performed to mitigate the immortal time bias. Table 2 lists the landmark points and the corresponding ORs. Later landmark points yielded smaller OR, indicating that without considering immortal time bias, the association between irAEs and response was overestimated.

Patients with early-onset irAEs had a RR of 50.0%, while those with late-onset irAEs had a RR of 81.8%. Similarly, patients with early-onset irAEs had a DCR of 60.0%, compared to 90.9% for those with late-onset irAEs.

The DCR in patients who experienced irAEs and those who did not were 76.2% and 51.0%, respectively. OR for disease control was 3.07 in the naïve analysis, and ORs in landmark analyses were lower than those in the former analysis.

## Progression-free survival and irAEs

Fig 1 shows PFS according to the development of irAEs. In the naïve analysis, the 1-year PFS rates were 47.6% and 21.7% for irAE+ and irAE- patients (p = 0.008). A similar result was

**Table 2. Odds ratios for objective response.**

| | Response | | | Disease control | | | No. of patients |
|---|---|---|---|---|---|---|---|
| | OR | 95% CI | p value | OR | 95% CI | p value | |
| Naïve | 3.94 | 1.45–10.70 | 0.005* | 3.07 | 1.04–9.04 | 0.042* | 119 |
| 4-week landmark | 3.76 | 1.38–10.22 | 0.007* | 2.88 | 0.98–8.50 | 0.055 | 116 |
| 8-week landmark | 3.58 | 1.31–9.74 | 0.013* | 2.69 | 0.91–7.95 | 0.074 | 113 |
| 12-week landmark | 3.03 | 1.11–8.30 | 0.031* | 2.11 | 0.71–6.32 | 0.181 | 104 |
| 16-week landmark | 2.85 | 1.04–7.83 | 0.042* | 1.92 | 0.64–5.78 | 0.246 | 101 |
| 20-week landmark | 2.61 | 0.95–7.19 | 0.064 | 1.66 | 0.55–5.05 | 0.369 | 97 |
| 24-week landmark | 2.92 | 1.01–8.45 | 0.049* | 1.88 | 0.56–6.25 | 0.305 | 93 |
| 28-week landmark | 2.86 | 0.98–8.32 | 0.054 | 1.75 | 0.52–5.87 | 0.365 | 91 |
| 32-week landmark | 2.63 | 0.90–7.69 | 0.076 | 1.5 | 0.44–5.09 | 0.516 | 88 |

Abbreviations: CI, confidence interval, OR, Odds ratio

* $p < 0.05$

observed in the 12 week-landmark analysis (1-year PFS rate of 47.6% for irAE+ patients and 27.2% for irAE- patients, p = 0.049). S1 Table presents the landmark points along with the corresponding 1-year PFS rates. Later landmark points showed higher PFS rates in irAE- patients, resulting in lower statistical significance.

PFS was comparable between patients with early-onset and late-onset irAEs (p = 0.159).

The univariate PH analysis revealed an association between irAEs and PFS (Table 3). However, irAEs were not associated with PFS in the EC analysis (HR, 0.77; 95% CI 0.41–1.42). Table 4 shows the results of the multivariable analyses. Adjusted HRs for irAEs were 0.49 (95% CI 0.28–0.85) in the PH analysis and 0.75 (95% CI 0.40–1.40) in the EC analysis. Thus, irAEs were not independent prognostic factors for PFS when immortal time bias was considered.

## Overall survival and irAEs

Fig 2 shows OS according to the development of irAEs. In the naïve analysis, the 1-year OS rates were 85.7% and 53.1% for irAE+ and irAE- patients, respectively (p< 0.001). In the 12 week-landmark analysis, the 1-year OS rates for irAE+ and irAE- patients were 85.7% and 66.5%, respectively (p = 0.006). S1 Table presents the landmark points along with the corresponding 1-year OS rates. Later landmark points demonstrated higher OS rates in irAE-patients.; however, the difference between irAE+ and irAE- patients remained consistently significant.

OS was comparable between patients with early-onset and late-onset irAEs (p = 0.380).

Univariate HRs for irAEs were 0.36 (95% CI 0.19–0.66) in the PH analysis and 0.51 (95% CI 0.27–0.95) in the EC analysis (Table 3). In the multivariable analyses, irAE was an independent prognostic factor for OS irrespective of analyses (adjusted HR of 0.38, 95% CI 0.20–0.71 in the PH analysis and adjusted HR of 0.50, 95%CI 0.26–0.95 in the EC analysis). Although irAEs were independently associated with OS in both analyses, the association became weaker when immortal bias was considered.

## Discussion

This study examined the association between irAEs and ICI treatment outcomes. We used landmark and EC analyses to address immortal time bias. We demonstrated that irAEs were

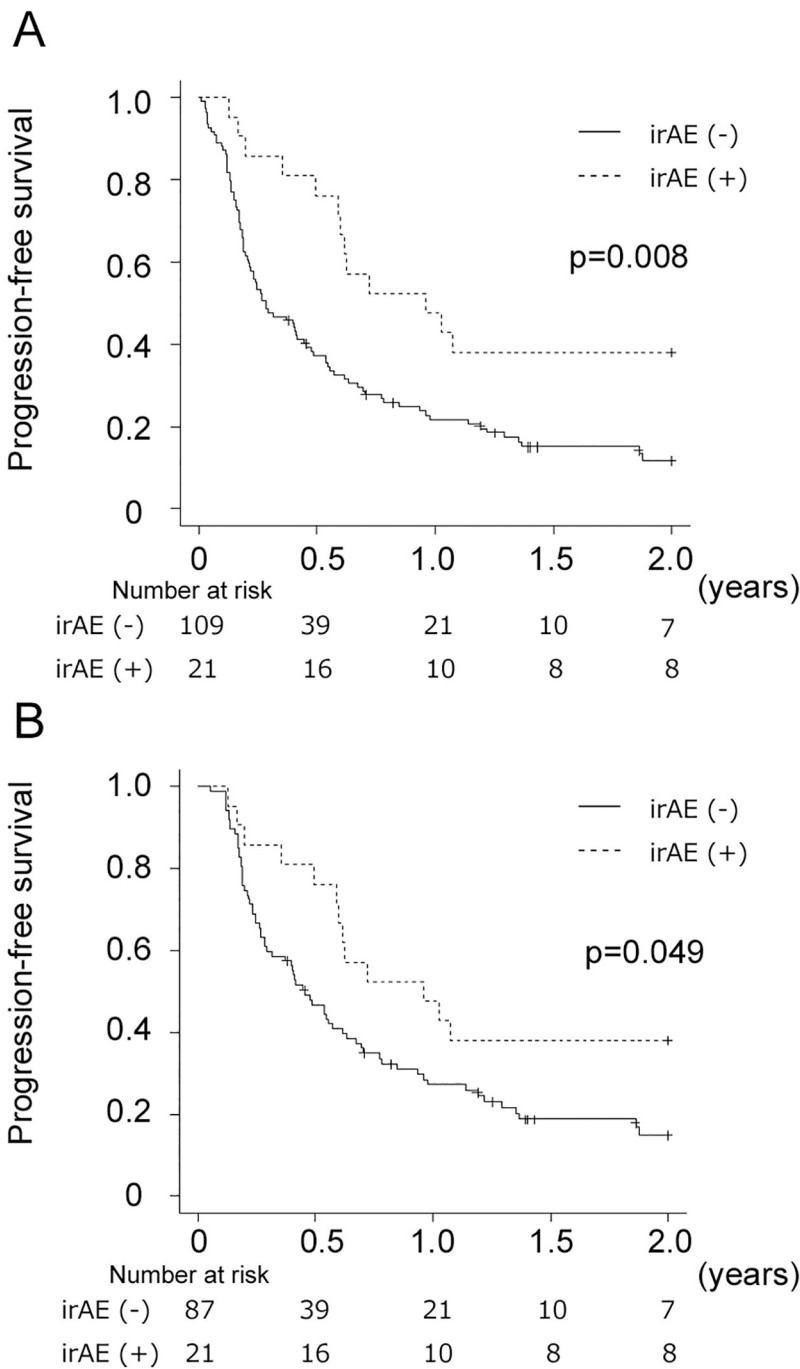

**Fig 1. Kaplan–Meier curves for progression-free survival according to irAE.** (A), a naïve analysis. (B), a 12-week landmark analysis.

independent prognostic factors for OS and found that the association between irAEs and treatment response was overestimated in analyses that did not account for immortal bias.

ICIs are part of the standard of care for various types of cancer, particularly R/M disease. ICIs are effective in only a small fraction of patients. In addition, ICI therapy is expensive

**Table 3. Univariate analyses for survival.**

| Clinicopathologic variable | Progression-free survival | | | Overall survival | | |
|---|---|---|---|---|---|---|
| | HR | 95% CI | p value | HR | 95% CI | p value |
| Age | | | | | | |
| 65 or more vs under 65 | 1.09 | 0.74–1.62 | 0.656 | 1.43 | 0.94–2.17 | 0.097 |
| Sex | | | | | | |
| Male vs female | 1.48 | 0.87–2.51 | 0.148 | 1.52 | 0.89–2.62 | 0.128 |
| Performance status | | | | | | |
| 1 vs 0 | 1.5 | 0.99–2.26 | 0.055 | 1.88 | 1.21–2.93 | 0.005* |
| 2 or more vs 0 | 1.93 | 1.11–3.38 | 0.021* | 3.04 | 1.71–5.39 | <0.001* |
| HPV status | | | | | | |
| Positive vs negative | 0.61 | 0.25–1.49 | 0.273 | 0.61 | 0.23–1.67 | 0.341 |
| Chemotherapy | | | | | | |
| Yes vs no | 0.72 | 0.47–1.10 | 0.133 | 0.66 | 0.42–1.05 | 0.078 |
| IrAE (Cox proportional hazard model) | | | | | | |
| Yes vs no | 0.49 | 0.29–0.84 | 0.010* | 0.36 | 0.19–0.66 | 0.002* |
| IrAE (Extended Cox model) | | | | | | |
| Yes vs no | 0.77 | 0.41–1.42 | 0.399 | 0.51 | 0.27–0.95 | 0.033* |

Abbreviations: IrAE, immune-related adverse event, CI, confidence interval, HPV, human papillomavirus, HR, hazard ratio

* p<0.05

and requires regular hospital visits. Therefore, biomarkers have been sought to identify patients suitable for ICI therapy. The most commonly used biomarker for PD-1 inhibition is PD-L1 expression in tumor specimens. However, the CheckMate 141 study on nivolumab for HNSCC did not demonstrate the usefulness of PD-L1 expression as a biomarker [8]. The KEYNOTE-048 study on pembrolizumab for HNSCC showed that pembrolizumab was effective regardless of PD-L1 expression [9]. Thus, PD-L1 expression is not a predictive biomarker for PD-1 inhibition in HNSCC. Tumor mutation burden is an established predictive biomarker for solid cancers treated with ICIs [2]. However, the predictive accuracy remains unsatisfactory. Other indices, such as the neutrophil-to-lymphocyte ratio (NLR) [10], sarcopenia [11], body composition indices [12], and nutritional indices [13] are prognostic factors for ICI therapy. Composite indices that combine several biomarkers, including patient factors, tumor factors, and blood markers, have been developed to more accurately predict prognosis [14]. Some of these have been validated in patients with HNSCC treated with ICIs,

**Table 4. Multivariable analyses for survival.**

| Clinicopathologic variable | Progression-free survival | | | Overall survival | | |
|---|---|---|---|---|---|---|
| | HR | 95% CI | p value | HR | 95% CI | p value |
| IrAE (Cox proportional hazard model) | | | | | | |
| Yes vs no | 0.49 | 0.28–0.85 | 0.011* | 0.38 | 0.20–0.71 | 0.002* |
| IrAE (Extended Cox model) | | | | | | |
| Yes vs no | 0.75 | 0.40–1.40 | 0.359 | 0.50 | 0.26–0.95 | 0.033* |

Abbreviations: IrAE, immune-related adverse event, CI, confidence interval, HPV, human papillomavirus, HR, hazard ratio

HRs were adjusted with age, chemotherapy, HPV status, and PS.

* p<0.05

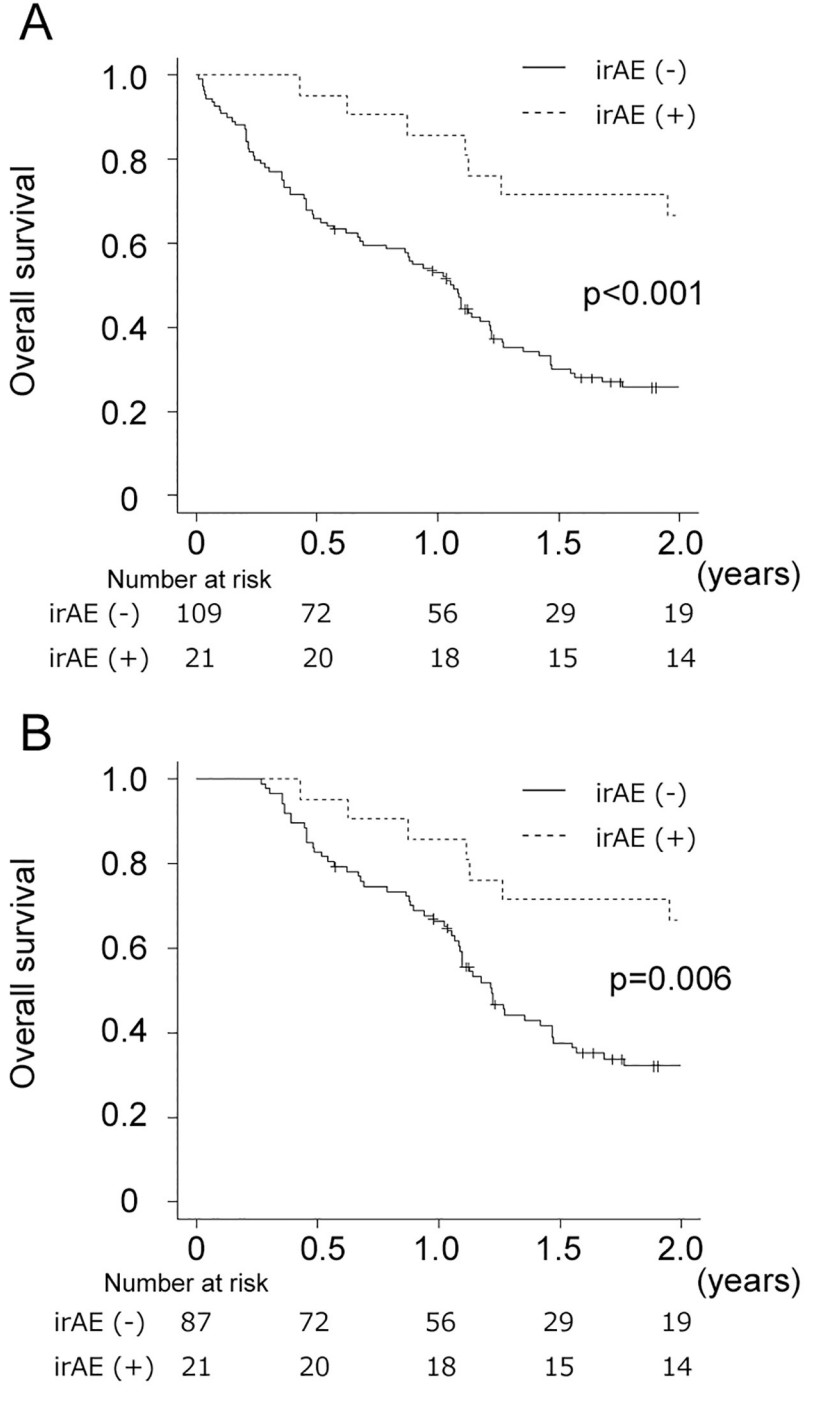

**Fig 2. Kaplan–Meier curves for overall survival according to irAE.** (A), a naïve analysis. (B), a landmark analysis.

and showed a strong association with OS and PFS [14]. However, these composite indices are not predictive factors, and are thus not used for treatment selection. Put together, no method exists currently for selecting patients suitable for ICI treatment. Owing to this, oncologists initiate ICI therapy and decide whether to continue or discontinue after

monitoring patient response. This decision is typically based on imaging studies. The objective response on imaging is evaluated according to the RECIST or iRECIST guidelines [7, 15]. In addition to imaging-based decisions, on-treatment biomarkers such as changes in serum LDH and NLR [16] may help determine continuation of treatment. One on-treatment maker of particular interest is the development of irAEs, which some clinicians consider a sign of ICI effectiveness. In patients with melanoma, anti-melanoma immunity targets not only melanoma cells, but also healthy melanocytes, resulting in vitiligo. Therefore, vitiligo, a skin irAE, is a good predictor of response in patients with melanoma [17]. However, our results indicated that irAEs in patients with HNSCC should not be considered as on-treatment marker of a good response. Previous studies have also explored the relationship between irAEs and ICI outcomes. A meta-analysis of irAE and ICI response was conducted for solid cancers [4], and of the 29 articles included in the study, only seven considered immortal bias, which can skew the results. For head and neck cancer, Foster et al. used landmark analysis to examine the relationship between irAEs and survival in patients treated with ICIs [5]. They concluded that the development of irAEs was a strong predictor of ICI outcomes. However, the study had a large number of patients excluded by landmark analysis, which reduced its statistical power. Moreover, no analysis of time-dependent variables was performed. Herein, we employed an EC model, which allows for the inclusion of onset timing as a time-dependent variable. In contrast, a traditional Cox model has a limitation in addressing the specific timing of irAE occurrence. We also performed a sensitivity analysis using differential time points in the landmark analysis. These analyses made our results scientifically sound compared with those of previous research. Based on our results, the onset of irAEs has limited value in predicting the efficacy of ICIs.

Our study had several limitations. First, we classified AEs requiring corticosteroid treatment and endocrine AEs necessitating hormone replacement therapy as irAEs. As a result, mild irAEs that resolve spontaneously may have been overlooked. This may have caused the low irAE incidence in our study. Furthermore, distinguishing whether these mild AEs are truly immune-related is often challenging, and accurately determining the timing of irAE onset can also be difficult. These factors may have influenced our findings. Second, it was a retrospective, single institutional study. Progression and irAEs were determined retrospectively using a chart review because of the lack of pre-specified criteria. To address the resulting bias, we conducted multivariable analyses and adjusted for confounding factors. Third, the number of patients was not large enough to provide sufficient statistical power. Landmark analyses further reduced the number of patients analyzed. Therefore, we could not conduct a multivariate analysis of response or disease control. Moreover, the small sample size led to increased variance and wide confidence intervals, making it difficult to achieve statistical significance. To overcome these limitations, a larger, prospective study—ideally multi-institutional—would be necessary.

In conclusion, irAEs were identified as independent prognostic factors for OS but not for PFS. However, when immortal time bias is not accounted for, the association between irAEs and oncological outcomes in patients with HNSCC treated with ICIs tends to be overestimated. Our findings highlight the importance of accurately assessing the true prognostic impact of irAEs, as immortal time bias can exaggerate their perceived benefits. While irAEs may lead to life-threatening conditions, their occurrence offers limited predictive value for deciding whether to continue ICI treatment. Therefore, clinicians should not base decisions about ICI continuation on the presence of irAEs alone. Future research should explore more thoroughly the role of irAEs as potential on-treatment markers for ICI outcomes, ensuring that analyses rigorously account for immortal time bias to provide accurate and reliable conclusions.

## Supporting information

**S1 Table. Landmark analyses for survival.**
(XLSX)

**S1 File. Patient raw data.**
(TXT)

## Author Contributions

**Conceptualization:** Yukinori Takenaka.

**Data curation:** Kiyohito Hosokawa.

**Formal analysis:** Koichi Tamura, Yukinori Takenaka.

**Funding acquisition:** Yukinori Takenaka.

**Supervision:** Hidenori Inohara.

**Writing – original draft:** Yukinori Takenaka.

**Writing – review & editing:** Koichi Tamura, Kiyohito Hosokawa, Takashi Sato, Takeshi Tsuda, Hirotaka Eguchi, Masami Suzuki, Takahito Fukusumi, Motoyuki Suzuki.

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
