## [Decision Letter · Decision Letter 0]

11 Sep 2024

PONE-D-24-31279Effect of immortal time bias on the association between immune-related adverse events and oncological outcomes following immune checkpoint inhibitors therapy for head and neck squamous cell carcinomaPLOS ONE

Dear Dr. Takenaka,

Thank you for submitting your manuscript to PLOS ONE. After careful consideration, we feel that it has merit but does not fully meet PLOS ONE’s publication criteria as it currently stands. Therefore, we invite you to submit a revised version of the manuscript that addresses the points raised during the review process.

Reviewers raised very constructive suggestions about more clarification of statistical analysis, interpretiaon, conclusion, and future perspectives. 

We look forward to receiving your revised manuscript.

Kind regards,

Hyun-Sung Lee, M.D., Ph.D.

Academic Editor

PLOS ONE

“This work was supported by JSPS KAKENHI; Grant Number JP24K12647.”

“This work was supported by JSPS KAKENHI; Grant Number JP24K12647.

The funder had no role in study design, data collection, analysis, decision to publish, or preparation of the manuscript.”

4. We note that there is identifying data in the Supporting Information file < renamed_52b73.txt>. Due to the inclusion of these potentially identifying data, we have removed this file from your file inventory. Prior to sharing human research participant data, authors should consult with an ethics committee to ensure data are shared in accordance with participant consent and all applicable local laws.

-Location data

Reviewers' comments:

Reviewer's Responses to Questions

**Comments to the Author**

1. Is the manuscript technically sound, and do the data support the conclusions?

Reviewer #1: Partly

Reviewer #2: Yes

2. Has the statistical analysis been performed appropriately and rigorously? 

Reviewer #1: Yes

Reviewer #2: Yes

3. Have the authors made all data underlying the findings in their manuscript fully available?

Reviewer #1: Yes

Reviewer #2: Yes

4. Is the manuscript presented in an intelligible fashion and written in standard English?

Reviewer #1: Yes

Reviewer #2: Yes

5. Review Comments to the Author

Reviewer #1: Tamura et al. investigated the association between immune-related adverse events (irAEs) and treatment outcomes in patients with head and neck squamous cell carcinoma (HNSCC) treated with immune checkpoint inhibitors (ICIs) and/or chemotherapies, focusing on the impact of immortal time bias. This retrospective study found significant differences in progression-free survival (PFS) and overall survival (OS) between patients with and without irAEs, demonstrating that irAEs might have a limited value as an independent prognostic factor for OS but not PFS when correcting for immortal time bias. The study concluded that the impact of irAEs on treatment outcomes was overestimated without considering immortal time bias and highlighted the need for clinicians to weigh the risks and benefits of ICI treatment carefully.

Despite its contributions, this manuscript has several shortcomings and weaknesses.

1) The novelty of this work is questioned, as previous studies have addressed immortal time bias in various cancer types. Although this study incorporates immortal time bias in its analysis, the findings do not significantly alter the previously understood associations in HNSCC and other cancers (Dall'Olio FG et al., 2021; Foster CC et al., 2021). It is essential to more clearly articulate the unique contributions and importance of this study.

2) This study included a small sample size for the naïve analysis as its limitation. In addition, this study mentions the reduced sample size due to landmark analyses, which limits the statistical power of the findings. However, it does not provide a comprehensive assessment of how these limitations might impact the validity of the results. A more detailed discussion of the potential biases introduced by the small sample size and mitigation strategies would be necessary to strengthen the conclusions.

3) While the authors have employed an extended Cox model to account for time-dependent variables, the manuscript does not fully explore how the timing of irAE onset may differentially impact patient outcomes. A more detailed analysis of the temporal aspects of irAE development, including the possible delayed effects on survival and progression, is recommended to add depth to the findings.

4) The choice of using landmark analysis and EC models is appropriate, given the study’s objectives. However, the manuscript could benefit from a more explicit rationale for these choices compared to other possible methods, such as incorporating time-varying covariates in a traditional Cox model. In addition, the rationale for selecting specific landmark time points is not well-explained and requires further elaboration to enhance clarity.

5) The conclusion should clearly summarize the key findings and their implications without ambiguity. It should include a clear message of how immortal time bias affects the association between irAEs and oncologic outcomes, how clinicians should interpret this association to balance the risks and benefits of ICI therapy in clinical settings, and what alternative strategies could be considered.

6) The manuscript suggested further research to investigate the magnitude of irAEs associated with ICI outcomes. However, it would benefit from more specific recommendations for future studies, such as prospective designs, larger sample sizes, or the use of more comprehensive biomarkers.

7) This study, being retrospective and from a single institute, has inherent limitations. It is recommended to describe how the author attempted to overcome these limitations without validation cohorts.

8) The study included 130 patients but analyzed 119 for an objective response. It should explain why 11 patients were excluded from the analysis. Also, Figures 1B and 2B included 108 patients for a survival analysis with a landmark analysis. Please clarify the patient number for each landmark analysis in Table 2.

9) In Table 1, 38 patients received the ICI therapy as 2nd or later lines. It is recommended to provide the details of previous treatment in these patients.

10) In Table 1, 38 patients received the ICI therapy with chemotherapy (fluorouracil + cisplatin). In these patients, it is recommended to describe how irAEs and AEs related to chemotherapy were differentiated and add the details of the irAEs and AEs.

11) Please update Table 1 to include all patient characteristics for both irAE (+) and irAE (-) groups, with the comparative statistics between the 2 groups.

12) irAEs occurred in 16.1% of patients, which is in the lower range compared to other ICI studies. Detailed information on any grade of irAEs or over grade 3 irAEs in each response criteria (CR, PR, SD, and PD), treatment for irAEs, and possible causes of low irAE incidence should be provided. Also, the association between the severe irAE (+) and oncologic outcomes should be added.

13) In Table 2, the p-value in the 20-week landmark analysis for response is 0.064 with an asterisk (*), indicating statistical significance based on the legend of Table 2 (* p<0.05). It should explain why this is statistically significant, or if this is an error, please correct it.

14) In Figures 1 and 2, it is recommended to add the p-value in each graph and the x-axis legend. Also, based on the median F/U period of 33.4 months, the authors should consider the extension of the x-axis.

15) In the Discussion section, the authors discussed predictive factors of ICI treatment, including PD-L1 expression. Basically, irAEs cannot be used to select patients or predict responses before starting treatment. This discussion is unfocused and irrelevant to their investigation. It is recommended to describe the relevant points for discussion.

16) In Table 4, the correct terminology would be “multivariable analysis” rather than “multivariate analysis,” as this analysis focused on understanding how multiple predictors affect single outcomes and used Cox proportional hazard models to evaluate the influence of these variables on survival outcomes. If the authors want to keep the original terms, the relevant statistics and rationale should be provided.

These modifications will enhance the manuscript’s clarity and the reliability of its conclusions, aligning it with standard reporting practices in oncological research.

Reviewer #2: The present paper investigates the association between immune-related adverse events (irAEs) and survival in head and neck cancer patients treated with immune checkpoint inhibitors (ICIs), while accounting for immortal time bias. This topic is highly relevant, as previous studies have demonstrated this association but often did not adequately adjust for this important bias.

The study is well-conceived, with a robust statistical analysis and conclusions that are, overall, convincing. However, I have two comments for the authors.

First, I question the final sentence: “Therefore, the balance between the benefits and risks of ICI therapy must be carefully weighted in clinical settings.” The benefits and risks of ICI therapy are typically best summarized by findings from randomized controlled trials, not from exploratory studies like this one, which focus on predictors of efficacy.

The biological hypothesis suggesting that irAEs could serve as a surrogate marker in patients receiving immunotherapy is based on the idea that a stronger autoimmune response induced by ICIs may correlate with a more robust antitumor immune response. If irAEs are indeed part of the primary causal pathway in this context, their development could fully mediate the effect of ICIs on survival outcomes (Amoroso et al., ESMO Open, 2023). Accounting for immortal time bias may attenuate this association further at the individual patient level.

In my view, the primary conclusion of this study should emphasize that future research exploring the potential role of irAEs as surrogates for ICI efficacy must rigorously account for immortal time bias.

Second, a potential limitation of the study is that the authors rely exclusively on clinician-reported adverse events (AEs) classified as irAEs. It is well established that patients often report adverse symptoms earlier and more frequently than clinicians. Furthermore, distinguishing irAEs from AEs arising from other causes, particularly in cases where ICIs are combined with other therapies, can be challenging. This introduces the possibility of misclassification bias, which may have influenced the study's findings.

6. PLOS authors have the option to publish the peer review history of their article (what does this mean?). If published, this will include your full peer review and any attached files.

Reviewer #1: No

Reviewer #2: **Yes: **Vito Amoroso

---

## [Author Response · Author response to Decision Letter 0]

23 Oct 2024

We attached the response to reviewers separately

---

## [Decision Letter · Decision Letter 1]

7 Nov 2024

Effect of immortal time bias on the association between immune-related adverse events and oncological outcomes following immune checkpoint inhibitors therapy for head and neck squamous cell carcinoma

PONE-D-24-31279R1

Dear Dr. Takenaka,

We’re pleased to inform you that your manuscript has been judged scientifically suitable for publication and will be formally accepted for publication once it meets all outstanding technical requirements.

Kind regards,

Hyun-Sung Lee, M.D., Ph.D.

Academic Editor

PLOS ONE

Additional Editor Comments (optional):

Reviewers' comments:

Reviewer's Responses to Questions

**Comments to the Author**

1. If the authors have adequately addressed your comments raised in a previous round of review and you feel that this manuscript is now acceptable for publication, you may indicate that here to bypass the “Comments to the Author” section, enter your conflict of interest statement in the “Confidential to Editor” section, and submit your "Accept" recommendation.

Reviewer #1: All comments have been addressed

2. Is the manuscript technically sound, and do the data support the conclusions?

Reviewer #1: Yes

3. Has the statistical analysis been performed appropriately and rigorously? 

Reviewer #1: Yes

4. Have the authors made all data underlying the findings in their manuscript fully available?

Reviewer #1: Yes

5. Is the manuscript presented in an intelligible fashion and written in standard English?

Reviewer #1: Yes

6. Review Comments to the Author

Reviewer #1: The authors have responded to my comments well, providing clear and detailed explanations throughout.

7. PLOS authors have the option to publish the peer review history of their article (what does this mean?). If published, this will include your full peer review and any attached files.

Reviewer #1: No

---

## [Editor Report · Acceptance letter]

13 Nov 2024

PONE-D-24-31279R1 

PLOS ONE

Dear Dr. Takenaka, 

I'm pleased to inform you that your manuscript has been deemed suitable for publication in PLOS ONE. Congratulations! Your manuscript is now being handed over to our production team.

Kind regards, 

on behalf of

Dr. Hyun-Sung Lee 

Academic Editor

PLOS ONE